# Gypsum-Related Impact on Antibiotic-Loaded Composite Based on Highly Porous Hydroxyapatite—Advantages and Disadvantages

**DOI:** 10.3390/ijms242417178

**Published:** 2023-12-06

**Authors:** Justyna Zalewska, Vladyslav Vivcharenko, Anna Belcarz

**Affiliations:** 1Chair and Department of Biochemistry and Biotechnology, Medical University of Lublin, Chodzki 1, 20-093 Lublin, Poland; anna.belcarz@umlub.pl; 2Independent Unit of Tissue Engineering and Regenerative Medicine, Chair of Biomedical Sciences, Medical University of Lublin, Chodzki 1, 20-093 Lublin, Poland; vladyslav.vivcharenko@umlub.pl

**Keywords:** highly porous hydroxyapatite, calcium sulfate dehydrate, gentamicin, Ca^2+^ uptake compensation, osteoblasts, drug carrier

## Abstract

Highly porous hydroxyapatite is sometimes considered toxic and useless as a biomaterial for bone tissue regeneration because of the high adsorption of calcium and phosphate ions from cell culture media. This negatively affects the osteoblast’s growth in such ion-deprived media and suggests “false cytotoxicity” of tested hydroxyapatite. In our recent study, we showed that a small addition of calcium sulfate dihydrate (CSD) may compensate for this adsorption without a negative effect on other properties of hydroxyapatite-based biomaterials. This study was designed to verify whether such CSD-supplemented biomaterials may serve as antibiotic carriers. FTIR, roughness, mechanical strength analysis, drug release, hemocompatibility, cytotoxicity against human osteoblasts, and antibacterial activity were evaluated to characterize tested biomaterials. The results showed that the addition of 1.75% gypsum and gentamicin caused short-term calcium ion compensation in media incubated with the composite. The combination of both additives also increased antibacterial activity against bacteria representative of bone infections without affecting osteoblast proliferation, hemocompatibility, and mechanical parameters. Thus, gypsum and antibiotic supplementation may provide advanced functionality for bone-regeneration materials based on hydroxyapatite of a high surface area and increasingly high Ca^2+^ sorption capacity.

## 1. Introduction

Ceramics-based composites recently played an important role in the design of biomaterials for bone tissue regeneration. Using a combination of different compounds, the defects of a single component can be eliminated. For example, hydroxyapatite (HA) ceramics of very high specific surface area can exhibit different properties in their interaction with the ions important for bone cell metabolism. Some nanopowders were shown to release calcium ions, which are crucial for bone tissue regeneration [1,2]. Other types of nanopowders (and very porous granules), designed for defects of bone, exhibit a very high rate of Ca^2+^, Mg^2+^, and HPO_4_^2−^ ion adsorption [3].

This affects bone tissue regeneration in both positive and negative manners. The positive aspect of this phenomenon is related to increased bio-mineralization after the implantation due to hydroxyapatite bioactivity. Hydroxyapatite intensely adsorbs calcium and phosphate ions from surrounding liquid, which leads to the formation of biological apatite in ceramics formation, and the rate of this process depends on the specific surface area of ceramics (and thus the number of nucleation sites). However, the disappearance of calcium and phosphate ions leads to negative consequences because the reduced ion availability for cells that are responsible for bone tissue formation can limit the osteoblasts’ proliferation and differentiation [3]. Calcium ions are considered one of the most important ions for the metabolism of bone tissue cells because they regulate the function and differentiation of chondrocytes, osteoclasts, and osteoblast lineage cells [4]. They have been reported to activate the expression of bone-related proteins (osteopontin, sialoprotein), mediated by specific calcium-dependent channels and kinase [5]. Many studies have already reported the experimental and clinical benefits of Ca^2+^-enriched materials in the treatment of teeth [6].

The combination of HA of a high specific surface area with calcium sulfate dehydrate (CSD; gypsum) may be helpful to solve this problem. Gypsum, obtained by hydration of calcium hemisulfate (CaSO_4_•½H_2_O; CSH), dissolves relatively quickly, releasing calcium ions to the environment, and is bioactive and safe in biomedical applications. Some sources report that gypsum can reduce the differentiation of osteoblast-like cells (MG-63) while their viability and proliferation seem to be unaffected [7]. However, another article proves its beneficial effect on ALP activity, proliferation, and differentiation of osteoblastic cells, suggesting the distinct osteoinductive potential of CSD [8]. This observation was supported by the report of He and colleagues, who created a β-TCP scaffold coated with gypsum. It exhibited the ability to affect the microenvironment parameters (pH and Ca^2+^ concentration), promoted rapid osteogenic differentiation and proliferation of bone marrow MSCs, and induced bioactivity in vivo [9]. In another article, addition of gypsum to hydroxyapatite-based purge for the repair of alveolar bone sockets of rats was proved to reduce osteoclasts and simultaneously increase osteoblast numbers in the trabecular area [10]. Interestingly, recent report revealed that gypsum calcined at specific temperature exhibited some antibacterial activity per se [11].

Moreover, it has been used in bone replacement materials available on the market (e.g., Stimulan^®^, Biocomposites LTD, Keele, UK, and Osteoset^TM^, Wright Medical Technology, Inc., Memphis, TN, USA [12]). There are reports that hydroxyapatite-calcium sulfate tested in vivo showed beneficial healing effects without signs of inflammation and immunogenic response in rat models [13].

We recently added CSD in an appropriate quantity to highly porous hydroxyapatite and observed the gypsum-mediated compensation effect on the adsorption of calcium ions by the hydroxyapatite [14]. Thus, CSD-supplemented porous hydroxyapatite can be used for bone tissue regeneration without a negative impact on osteoblast metabolism.

However, bone tissue regeneration encounters other problems, namely, bacterial infection of implants and surrounding tissues. Healthy intact bone is relatively resistant to infection, but when a large inoculum of bacteria is introduced, from trauma, ischemia, or the presence of foreign bodies like implants, it becomes susceptible to disease. The appearance of bacterial biofilm is known to be the more dangerous because of the phenotypic resistance of biofilm-entrapped bacteria to antibiotic therapy and the related persistence of bone infections [15]. Hydroxyapatite does not exhibit any antibacterial activity. Its surface architecture may support the adhesion of bacteria and therefore can even be used to separate and concentrate bacteria from foods [16]. This biomedical deficiency of HA is often thought to be eliminated by various post-treatment modifications [17]. The presence of CSD in HA-gypsum composites may adversely affect the antibacterial properties of the composite because calcium ions released by gypsum are likely to support not only osteoblast but also bacterial growth. Calcium sulfate was found long ago to stimulate some legume bacteria [18]. There is growing evidence that calcium ions serve as a determinant of growth, differentiation, motility, a host environment for invading bacteria, and regulate the main events of host colonization and bacterial virulence [19,20]. It was even found for *Bacillus subtilis* that subpopulations of biofilm-residing bacteria, so-called mineral-forming cells, are essential for biofilm formation [21]. Thus, CSD’s presence in composites designed for bone tissue regeneration is likely to support bacterial growth similarly to osteoblast growth. We hypothesize that supplementation of porous HA-CSD composites with antibacterial agents such as antibiotics, noble metal ions, nanoparticles, and others can prevent this undesirable phenomenon and eliminate the disadvantages of the proposed composite. This strategy was already shown in a study by Butini et al., who reported that a gentamicin-supplemented calcium sulfate-hydroxyapatite bone graft substitute showed high antibacterial activity [22]. Similarly, bioactive glass + calcium sulfate material loaded with antibiotics exhibited high efficacy in the treatment of bacterial infection developed in laboratory rabbits treated with the biomaterial [23]. Gentamicin, one of the antibiotics nowadays the most frequently used in the treatment of osteomyelitis [24,25], was selected for this study.

To verify the above hypothesis, we synthesized the composite based on highly porous granular HA (of high capacity to adsorb Ca^2+^, Mg^2+^, and HPO_4_^2−^ ions), CSD particles, and polymeric β-1,3-glucan (curdlan) as a matrix. The resulting composite was supplemented with gentamicin as an antibacterial agent. The effect of the produced biomaterial on human osteoblasts behavior (cell viability and proliferation), bacterial growth and adhesion, human blood hemolysis, and blood clot formation, as well as some mechanical features of the composites, were determined. In general, it seems that the combination of CSD and gentamicin enhanced the positive properties of the composite based on highly porous hydroxyapatite designed for biomedical purposes.

## 2. Results and Discussion

### 2.1. Composite Characterization

Composites based on hydroxyapatite of a very high surface area, prepared with the addition of CSD and gentamicin, were designed to compensate for ion adsorption caused by HA to achieve two goals: (i) promote osteoblast adhesion and proliferation and (ii) to protect against bacterial infection. The choice of CSD as one of the additives was based on the results of our previous article [14]. The additives should also be neutral for the physicochemical properties of the basic composite.

Synthesized composites were characterized using several techniques. FTIR analysis, which reflects the chemical composition of materials, revealed that the HA signal dominated over other compounds in all samples. The bands characteristic for PO_4_^−3^ groups were observed: ν_1_ at 963 cm^−1^, ν_3_ at 1022 cm^−1^ and 1088 cm^−1^, and ν_4_ at 562 cm^−1^ and 599 cm^−1^. The presence of –OH groups typical for HA was indicated in the composite spectra by bands located at 3572 cm^−1^ and 630 cm^−1^. Gentamicin and CSD were not detected in the composites, probably due to their low content (less than 5% of the total sample composition). Curdlan’s presence in the composites was indicated by a very small shift of the spectra around 966 cm^−1^ (C_1_-O-C_3_ stretching vibration, characteristic for β configuration) [26]. The values of the curdlan-to-phosphate ratio in all composites increased to 0.030–0.032 in comparison with 0.020 for pure HA (Figure 1a). No additional band shifts were observed in comparison with reference spectra of HAP, CSD, CR, and gentamicin, suggesting the lack of chemical interactions between the compounds of the composites (Figure 1a). This observation remains in agreement with our previous report on the HAP-CR composite [26].

Composites were subjected to compression after complete soaking in phosphate buffer. Stress–strain curves (Figure 1b) and images of samples after compression (Figure 1c) showed that they were viscoelastic and that compression energy diffused perpendicularly. No significant differences were found between the reference and additives-containing samples (Figure 1d). Thus, CSD and gentamicin addition did not affect the mechanical properties of the composites.

Sample roughness was evaluated using 3D Scanning Laser Microscope LEXT™ OLS5100. In the case of tested biomaterials with amorphous surfaces, surface area roughness was measured. The dimension of the measured sample surface area was 1281 × 1280 μm, and analysis was conducted using 211× magnification. The deepest valleys on the sample surface possess a blue color, whereas sample peaks possess a red color, as can be seen in Figure 2.

According to the conducted evaluation, sample C + CSD revealed the highest surface area roughness (62.5 ± 7 µm). All modifications slightly increased surface roughness compared to the curdlan biomaterial; nevertheless, no statistically significant differences between samples were noted.

### 2.2. Biocompatibility

The ion adsorption test showed that the curdlan-HA composite (C), as well as the gentamicin-enriched one (C + G), caused a significant uptake of ions from the culture medium (Figure 3). The addition of CSD inhibited this uptake (for Ca^2+^ and Mg^2+^) while increasing the trend to adsorb phosphate ions. However, this effect was the most significant during the first day of the experiment and decreased with time (Figure 3). The results are similar to those reported earlier and confirm that CSD shows a short-lasting effect reversing the ion adsorption caused by HA of a high specific surface area [14].

Gentamicin release profiles for C + G and C + CSD + G composites were very similar (Figure 4). Slightly more drug (46.2% of the total loaded drug) was released from the gypsum-containing biomaterial than the gypsum-free one (42.4 of the total loaded drug). This may suggest that soluble gypsum may slightly facilitate drug release by pore formation in the matrix [14]. The drug was released for 5 h; the remaining part was stably maintained within the composite structure. We may hypothesize that this drug amount will protect the composite after the implantation and perhaps be slowly released due to osteoclast-mediated digestion. The Korsmeyer–Peppas model used for release mode determination suggested a pore-dependent (Fickian) release type (R^2^ value of approx. 0.98 showed good data fitting to the model) [27].

Besides relevant physicochemical and microstructural properties, biomaterials dedicated to bone implant application should exert appropriate biological characteristics supporting and accelerating bone tissue regeneration. Biocompatible biomaterials reveal a proper cellular response without any toxicity to host tissue [28]. In the present work, curdlan-based scaffolds modified with calcium sulfate dehydrate (CSD) and/or gentamicin (G) were subjected to biocompatibility analysis. Cytotoxicity of produced samples was estimated using the indirect MTT test following the ISO 10993-5:2009 standard [29]. According to the obtained results, fabricated materials were non-toxic due to the high viability of MC3T3-E1 cells compared to the negative control. A statistically significant reduction in cell viability was noted for the curdlan scaffold modified with gentamicin; nevertheless, it exceeded 82%, indicating non-toxicity (according to ISO standards, a toxic sample is characterized by cell viability lower than 70%) (Figure 5a). However, the addition of CSD (for C + CSD + G samples) reversed this trend. The results also showed that a properly selected dose of gypsum may be safe for osteoblasts, although CSD in excess may reveal toxic effects on osteoblasts metabolism [30]. Our observations are therefore in agreement with those of Ghorbanzadeh et al., suggesting the beneficial effects of a mixture of HA of high specific surface area and calcium sulfate on the metabolism of bone cells [31].

The direct live/dead cytotoxicity assay confirmed the lack of a cytotoxic effect (Figure 5b). Mouse preosteoblasts seeded on biomaterials and cultured for 2 days possessed green fluorescence with flattened morphology (Figure 6). Single cells stained red were noted, indicating high biocompatibility and non-toxicity of the tested biomaterials. Worth noting is the fact that the gentamicin-modified sample did not reveal a higher amount of dead (stained red) cells compared to other samples. The obtained result may indicate a decrease in cell metabolism in the case of the C + G sample, which is reflected in the result obtained in the MTT test. The number of mouse primary calvarial preosteoblasts on the surface of fabricated scaffolds was determined 1, 3, and 7 days after seeding using total LDH analysis (Figure 5b). The performed study revealed that all fabricated biomaterials supported cell proliferation. All samples revealed an increase in cell number over time, indicating high biomaterial biocompatibility. After the seventh day of cell culture, the sample modified with CSG and gentamicin revealed the highest amount of MC3T3-E1 cells. The introduced modifications to the curdlan material did not cause a cytotoxic effect or decrease the biocompatibility of the sample. The best quantitative result for cell growth was obtained for curdlan material combined with CSG and gentamicin.

Contact with blood is important for implantable materials because it may affect their biological safety. Regarding the tested samples, none caused hemolysis (Figure 7a). However, clot formation was slightly but statistically significantly inhibited by the presence of the C composite and gentamicin-enriched composites. This effect was not detected only for the C + CSD material, suggesting hemocompatibility of CSD supplementation (Figure 7b).

### 2.3. Antibacterial Activity

The most important feature of antibiotic-loaded biomaterials is their antibacterial activity. Therefore, this aspect was studied for two bacterial strains that are the most frequent in orthopedic infections: *Staphylococcus aureus* (Gram-positive strain) and *Escherichia coli* (Gram-negative strain).

Semi-quantitative tests of growth inhibition zones created some pilot knowledge. As expected, no bacterial inhibition was observed around the composites without gentamicin, while distinct zones of growth inhibition were noted for gentamicin-loaded ones. Interestingly, higher antibacterial activity was detected for the biomaterial containing gypsum (27 mm and 31 mm) than for the biomaterial without CSD (17 mm), despite the same drug loading in both samples (Figure 8a). The presence of CSD itself did not justify this difference because the composite with CSD and without the drug did not inhibit bacterial growth. It is more likely that CSD, being more soluble than HAp, gradually dissolves in contact with the agar medium, thus creating pores in the composite structure that facilitate gentamicin release. This phenomenon was confirmed in our previous work, where we observed the appearance of pores in a CSD-containing multiphasic composite after incubation in SBF [14]. Thus, increased porosity due to the dissolving CSD might have facilitated the release of gentamicin.

The bacterial-killing activity of the composites was measured according to a standard adapted for porous materials [32]. In this test, bacteria are incubated within the porous material and then eluted from it, followed by plating on nutrient agar to evaluate CFU growth. Hypothetically, the amount of bacteria eluted from the neutral composite (C) should be similar to the inoculated control unless nonspecific bacterial adhesion to the biomaterial or bacterial growth stimulation by the biomaterial itself appears. This was indeed found for *S. aureus* but not for *E. coli* strain—in the latter case, approx. 2-fold more bacteria were eluted from the neutral composite (Figure 8b,c). These observations concerning *E. coli* growth stimulation would be logical if curdlan was digested by the bacteria and consumed. However, it was reported earlier that *E. coli* does not metabolize curdlan, even in its pre-digested form (β-(1→3)-oligosaccharides) [33]. Thus, we cannot explain this result observed for *E. coli*. The presence of CSD in the composite increased (approx. 4-fold) the growth rate of *S. aureus* (for C + CSD in comparison with C biomaterial), which can be explained by the stimulatory effect of calcium ions released by CSD. Calcium ions have been reported to affect the growth and metabolism of all organisms, including different bacteria, in a concentration-dependent manner [34,35,36]. This effect was not observed for *E. coli* (the viability of bacteria for C and C + CSD samples was similar). However, this effect might have been masked by the *E. coli* growth increase observed in the presence of the C composite.

In contrast, the complete lack of viable bacteria was noted for both composites containing gentamicin (Figure 8b,c). This confirms the high bactericidal effect of gentamicin on bacteria.

The adhesion of bacteria to biomaterials is potentially dangerous because it initiates biofilm formation and the accumulation of drug-resistant bacteria within. In our study, the adhesion of bacteria was estimated by confocal laser microscopy after staining with a live/dead stain kit. Representative images of composite samples after staining are presented in Figure 9a, where viable bacteria were stained green and dead ones red. According to our observation, only a single bacteria adhered to composite surfaces, and all bacteria adhered to composites were viable. Comparison of the area of green fluorescence on particular images showed the tendency common for both strains: the presence of CSD reduces bacterial adhesion in comparison with the control composite (Figure 9b). For drug-containing composites, the trend was less obvious. *S. aureus* adhesion was inhibited by the presence of gentamicin, and the cumulative anti-adhesive effect of CSD and gentamicin was notable. For *E. coli*, in contrast, the effect of gentamicin is less significant; moreover, no cumulative effect of gypsum and drug was observed. In general, the susceptibility of *E. coli* and *S. aureus* to gentamicin was comparable because gentamicin MICs for these two strains are similar [37,38]. Also, the growth inhibition zones around the composite with CSD and the drug (C + CSD + G) were smaller for *E. coli* than for *S. aureus* (Figure 8a). These observations may suggest that the tested drug-loaded composite is more effective against Gram-positive than Gram-negative bacteria. This also remains in agreement with reports of other scientific groups working with β-tricalcium phosphate/calcium sulfate (β-TCP/CS) bone graft substitute for compatibility with vancomycin in combination with tobramycin or gentamicin. Tested combinations were shown to inhibit the formation of bacterial biofilm and significantly reduce the bioburden of pre-grown biofilms together with osteogenesis stimulation [39].

In our opinion, the concept of the presented design contains two basic elements that may affect various aspects of bone tissue regeneration: antibiotic loading and gypsum supplementation. The porous formula of the composite allows it to absorb water from surrounding liquid. This results in the gradual dissolution of the gypsum phase (1.75% of total composite weight), releasing calcium ions. This compensates for the adsorption of calcium and phosphate ions from the medium driven by hydroxyapatite of high specific surface area and positively affects the growth and proliferation of osteoblasts, especially in the first adaptive phase of cell–biomaterial contact. Gentamicin undergoes the release from the matrix, inhibiting the growth of bacteria. It seems that the rate of drug release is facilitated by gypsum dissolution accompanied by the formation of additional pores within the composite. Drug release is higher (Figure 4) and zones of bacterial growth inhibition are larger (Figure 8a) in the case of gypsum-and-antibiotic-loaded material in comparison to antibiotics-loaded one. Although evaluation of the bacterial-killing potential of the materials suggested that calcium ions released from gypsum may somewhat stimulate bacterial growth, it seems that simultaneous loading with antibiotics eliminates this effect (Figure 8b,c). Notably, these additives are neutral for mechanical properties and the 3D structure of final composites (probably due to their low content). Overall, it seems clear that low doses of CSD and antibiotics, added to high-specific-area hydroxyapatite-based biomaterials, may be a promising tool for the application of this type of ceramics in the biomedical field.

## 3. Materials and Methods

### 3.1. HA Preparation

HA powder was synthesized by wet chemical precipitation method using Ca(OH)_2_ H_3_PO_4_ (1.67 Ca/P molar ratio). Briefly, to the beaker containing 1 M Ca(OH)_2_ solution, 1.67 M H_3_PO_4_ was added dropwise using a peristaltic pump (continuous stirring, RT). Next, the pH of the solution was adjusted to 11.0 using 2 M NaOH. Then, the solution was aged for 96 h. The liquid above the precipitate was then decanted, and the resulting precipitate was rinsed several times with distilled water to the neutral pH, dried at 90 °C, and calcined in a muffle furnace at 800 °C. The resulting ceramics were ground in a mortar and sieved to obtain granules ≤ 0.4 mm.

### 3.2. Composite Preparation

Four types of composites were used for the experiments (Table 1). The preparation of CSD from CSH (Sigma-Aldrich, Taufkirchen, Germany) was described in [14]. Curdlan was obtained from *Alcaligenes faecalis* (Fujifilm Chemicals, Osaka, Japan), specific rotation [A]^20^/_D_). The final concentration of gentamicin in the dry mass of the composite is 1 mg/g.

The components of the composite were thoroughly mixed, transferred to glass tubes with a diameter of 9 mm, heated for 15 min at 95 °C, cooled, and cut into 4–5 mm discs. Then, the obtained biomaterials were dried.

### 3.3. Composite Characterization

FTIR-ATR: IR spectra were collected using a Vertex 70 spectrometer equipped with an ATR-diamond crystal accessory (Bruker, Billerica, MA, USA), resolution 4 cm^−1^, and 64 scans per spectrum. Samples were dried before the analysis at 37 °C for 24 h to remove the unbound water. The presence of curdlan in composites was evaluated based on the ratios of curdlan-to-phosphate characteristic absorption peaks (966 cm^−1^/1022 cm^−1^) in comparison with the pure HAp infrared spectrum. The spectra were then analyzed using OPUS 7.0 software (Bruker, Billerica, MA, USA) to obtain the values of 966 cm^−1^ (curdlan) and 963–1088 cm^−1^ (phosphate) band intensity relative to the local baseline.

Biomaterial surface roughness: This was evaluated using a LEXT™ OLS5100 3D Scanning Laser Microscope (Olympus Corporation, Tokyo, Japan). Biomaterial area surface roughness measurements were calculated in seven different areas (dimension of measured area: 1280 × 1281 μm) of each sample using 211× magnification.

Mechanical parameters: Mechanical parameters were evaluated for composite discs soaked in 0.1 M phosphate buffer, pH 7.4 to mimic operating conditions, in triplicate. The compression test was carried out using the EZ Test EX-SX universal testing machine (Shimadzu, Kyoto, Japan) equipped with the Trapezium program and a force sensor of 100 N, with a crosshead rate of 5 mm/min, starting after obtaining a force value of 0.05 N to eliminate gaps between the sample and the grips. The mechanical compression was carried out until 30% compression was reached. The force detected at 30% graft compression was evaluated.

### 3.4. Evaluation of Ionic Reactivity of Composites in Culture Medium (DMEM/F12)

The composite discs were placed in a 6-well plate and sterilized using ethylene oxide for 1 h at 55 °C. Then, DMEM medium was added to each well (1 mL/0.1 g of dry composite). The plate was incubated for 1 week at 37 °C. Each day, the culture medium was withdrawn from the well, and a new portion of the medium was added. Then, the concentration of ions (Ca^2+^, Mg^2+^, and HPO_4_^2−^) was determined at specific time points (after the 1st, 3rd, and 7th day of the experiment) using the following commercial kits: Calcium CPC, Magnesium, and Phosphorus; (Biomaxima, Lublin, Poland) and Synergy H4 Hybrid Microplate Reader (Biotek, Winooski, VT, USA). The experiment was performed in triplicate, and the results were expressed as mean values ± SD (standard deviations).

### 3.5. Gentamicin Release

Drug release evaluation was performed in a closed-loop system. USP 4 CE1 Sotax (Donau Lab, Basel, Switzerland)) was used to test the amount of antibiotic released from 2 types of gentamicin-containing composites: C + G and C + CSD + G. For this purpose, 1 g of each composite containing 1 mg of gentamicin was placed in units. The composites were soaked with 40 mL of PBS pH 7.4, under laminar flow of 1 mL/min, at 37 °C. At specified time intervals, 1 mL of fluid was taken to estimate drug concentration, and the system was supplemented with 1 mL of fresh PBS to maintain a constant volume. Determination of gentamicin concentration in collected samples was carried out according to the procedure described in other publication (Ginalska et al., 2004). Antibiotic concentrations were calculated based on the results of 4 independent experiments (each in triplicate). Korsmeyer–Peppas and Higuchi models were used to describe the kinetics of drug release according to the general equation M_t_/M_∞_ = kt*n*, where M_t_ is the amount of drug released from the samples at time t, M_∞_ is the cumulative amount of gentamicin released over time t→∞, k and *n* is kinetic constant and release exponent, respectively. The mechanism of drug release was interpreted by non-linear regression analysis using the Statistica 10 program (TIBCO Software Inc., Palo Alto, CA, USA).

### 3.6. Cell Culture Experiments

Cytotoxicity assay and cell proliferation test were conducted using a mouse primary calvarial preosteoblast cell line (MC3T3-E1 Subclone 4, CRL-2593, ATCC-LGC standards, Teddington, UK). Mouse preosteoblasts were cultured in Alpha Minimum Essential Medium (GIBCO, Life Technologies, Carlsbad, CA, USA) supplemented with 10% fetal bovine serum (FBS, Pan-Biotech GmbH, Aidenbach, Bavaria, Germany), 0.1 mg/mL streptomycin, and 100 U/mL penicillin (Sigma-Aldrich Chemicals, Warsaw, Poland). MC3T3-E1 cells were cultured in an incubator at 37 °C in a humidified atmosphere of 5% CO_2_ and 95% air.

*Cytotoxicity evaluation*: Cytotoxicity of produced and sterilized using ethylene oxide biomaterials was tested indirectly using sample extracts and directly by seeding cells on the biomaterials and visualizing using Live/Dead Double Staining Kit (Sigma-Aldrich Chemicals, Poland). The experiment with biomaterial was carried out by ISO 10993-5:2009 standard [29]. In the indirect cytotoxicity evaluation, MC3T3-E1 cells seeded in 96-multiwell plates at a concentration of 2 × 10^5^ cells/mL and cultured for 1 day were treated with biomaterial extracts for 24 h. Cells treated with Alpha Minimum Essential Medium (MEM-α) served as a negative control. Afterward, cell viability was evaluated by colorimetric MTT test as described earlier [40] and total LDH test according to the manufacturer’s instructions. The obtained results were presented as the percentage of negative control. Conducted assays were repeated in three independent experiments.

In the direct cytotoxicity test, MC3T3-E1 cells at a concentration of 2 × 10^5^ cells/mL were cultured on biomaterials for 2 days, stained using calcein-AM and propidium iodide in accordance with manufacturer protocol, and visualized by a confocal laser scanning microscope (CLSM, Olympus Fluoview equipped with FV1000, Olympus Corporation, Tokyo, Japan). Cells cultured on a polystyrene surface served as a control.

Cell Proliferation Test: Cell proliferation on produced biomaterials was determined by culturing MC3T3-E1 cells for 1, 3, and 7 days. Biomaterials were stuck with agarose to the plate well, and 1 × 10^5^ cells were seeded on each sample. After the selected time intervals, cell number was determined using the total LDH test by measuring the activity of the lactate dehydrogenase after the cell lysis. The total LDH analysis was conducted according to the manufacturer’s instructions.

### 3.7. Hemolysis and Blood Clot Formation Tests

Human blood collected for citrate from a healthy volunteer was used for the experiment (with the consent of the Bioethics Committee of the Medical University No. KE-0254/258/2020). Plasma hemoglobin (Hb) was determined by reaction with Drabkin’s reagent and calibration curve (Synergy H4 hybrid microplate reader, Biotek, Winooski, VT, USA, and 96-well plates), and its concentration was 0.98 mg/mL (within the acceptable limit below 2 mg/mL).

Hemolysis test: composite discs (50 mg ± 2 mg) were immersed in 2 mL of blood (diluted 100× with PBS pH 7.4, free of Ca^2+^ and Mg^2+^ ions). The positive control was 0.1% Triton X-100, and the negative control was 50 mg ± 2 mg HDPE (high-density polyethylene, Sigma-Aldrich, St. Louis, MO, USA). Then, the discs were incubated for 3 h at 37 °C on a hematology stirrer CAT RM 5-30V (CAT, Ballrechten-Dotingen, Germany), 10 rpm. In the last stage of the experiment, the plasma level of Hb released from erythrocytes was determined.

Clot forming test (using Drabkin’s reagent) is a method based on the measurement of hemoglobin released from free blood cells. In the first step of the experiment, whole blood was activated with 10 mM CaCl_2_. A total of 200 µL of activated blood was then spotted onto 130 mg ± 3 mg composite or HDPE discs (positive control). Non-activated whole blood served as a negative control. The samples were incubated for 30 min at 37 °C without shaking. Then, all types of composites were incubated with 2.5 mL of distilled water for 5 min, after which time the liquid was collected and its hemoglobin content was measured using the reaction with Drabkin’s reagent, as described above. Each experiment was performed in triplicate. Statistically significant differences between the negative control and the various samples were accounted for at *p* < 0.0001, according to Dunnett’s one-way post hoc ANOVA (GraphPad Prism 9.4.0 software, San Diego, CA, USA).

### 3.8. Antibacterial Activity

Strains and Maintenance: Two bacterial strains (*Staphylococcus aureus* ATCC 25923, *Escherichia coli* ATCC 25922) were plated on agar plates (Mueller-Hinton Agar, Biomaxima, Lublin, Poland). The plates were incubated for 24 h at 37 °C. Then, the bacteria were scraped from the medium and bacterial inocula were prepared. Selected bacterial strains, according to literature data, are responsible for the majority of post-implantation infections in orthopedic departments.

Agar Plate Test: For this test, 4 types of composites were used. The agar was poured into Petri dishes, and after solidification, holes with a diameter of 7.5 mm were cut out. Composite discs (95 ± 5 mg) were placed in the holes and were wetted with sterile deionized water (150 µL). After the biomaterials were hydrated, an additional thin layer of agar was poured onto the plate. After the agar solidified, 350 μL of a bacterial solution with a concentration of 3.0 × 10^7^ CFU/mL was dropped on its surface. The plates were incubated for 24h at 37 °C, after which the zones of bacterial growth inhibition were measured.

Antibacterial activity test (based on the standard: AATCC Test Method 100–2004) [32]: Briefly, sterile composites (two without gentamicin and two with gentamicin) were placed in sterile Petri dishes and wetted as above (in triplicate). Then, two bacterial strains inoculates (1.5 × 10^7^ CFU/mL) were applied onto the composites in volume of 200 µL per disc. The composite discs were incubated at 37 °C for 24 h. A total of 200 µL of each inoculate incubated without the samples served as controls. Then, controls and samples were transferred to sterile 0.9% NaCl (15 mL) and vigorously shaken (1 min.) to elute the bacterial cells. Samples containing the eluted bacteria were plated on Mueller-Hinton agar plates using an automatic plater (EasySpiral Dilute; Interscience, Saint Nom la Bretêche, France). Plates were incubated at 37 °C for 18–24 h. CFU was counted using the Scan 300 colony counter (Interscience, Saint Nom la Bretêche, France).

Bacterial adhesion: The composite discs were incubated in suspensions of *S. aureus* and *E. coli* bacteria (1 mL approx. 1.0 × 10^8^ cells/ mL, in M-H broth), for 2 h, at 37 °C with gentle agitation (50 rpm; Innova, New Brunswick, NJ, USA). Then, the composites were rinsed gently with 0.9% NaCl (200 mL, 2 times). Washed samples were incubated with a Viability/Cytotoxicity Assay Kit for Bacteria Live & Dead Cells (Biotium, Fremont, CA, USA) in 0.9% NaCl according to manufacturer’s instructions. After staining, the samples were washed in 0.9% NaCl to remove non-absorbed dye. Adhered bacteria were visualized by confocal microscopy (Olympus Fluoview FV1000; Olympus, Tokyo, Japan) and ANOVA (GraphPad Prism 9.4.0 software, San Diego, CA, USA).

### 3.9. Statistical Analysis

Data analysis was performed using GraphPad Prism Software (version 9.4.0). One-way ANOVA followed by Tukey’s multiple comparison test was applied to determine statistically significant differences between the tested samples. The significance level was considered at *p* < 0.05. The results were expressed as the mean values ± standard deviation (SD).

## 4. Conclusions

According to our results, gypsum-enriched composites designed for bone tissue regeneration can serve as a matrix for antibiotic delivery to bone defect sites. The results showed that the addition of 1.75% gypsum and gentamicin caused short-term calcium ions compensation in media incubated with the composite. The combination of both additives also increased antibacterial activity for Gram-positive and Gram-negative strains representative of bone infections without negative effects on cell proliferation, hemocompatibility, and mechanical parameters. The properties of these two additives likely cooperate in the final effect on osteoblasts and bacteria. According to our observations, the dissolving gypsum may not only release calcium ions that compensate for their uptake from the environment by high-specific-area hydroxyapatite. It may also facilitate the action of antibiotics by pore generation in the biomaterial and enhancement in liquid circulation within its structure. Thus, gypsum and antibiotic supplementation may provide advanced functionality for bone-regeneration materials based on hydroxyapatite of a high surface area and increasingly high Ca^2+^-sorption capacity.

## Figures and Tables

**Figure 1 ijms-24-17178-f001:**
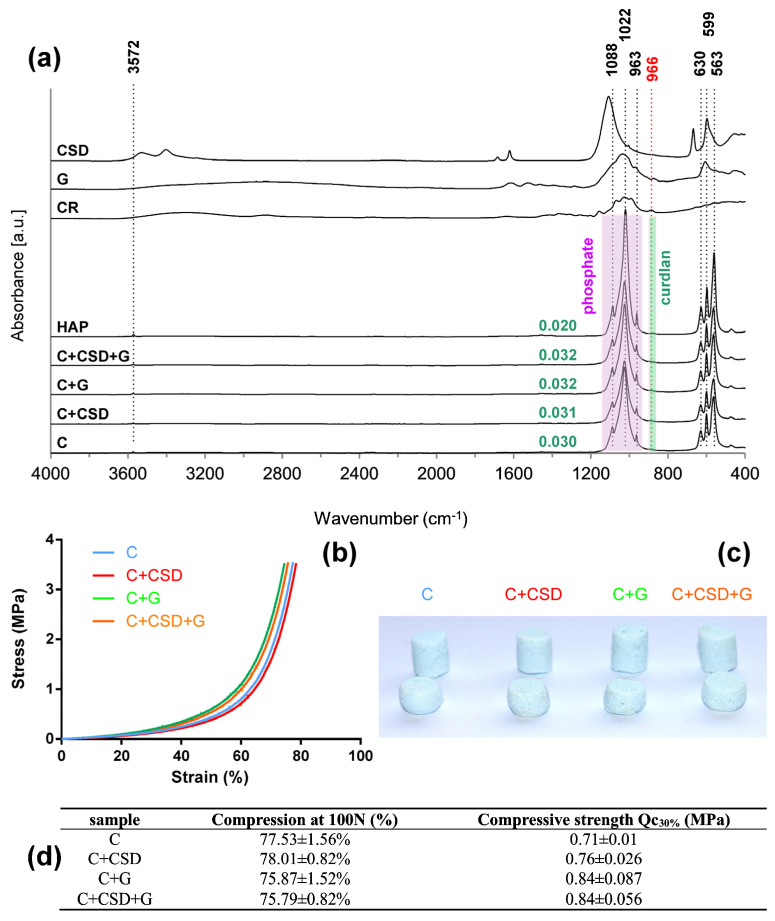
FTIR spectra of the composites and their compounds. Values in green denote the ratios of 966 cm^−1^ peak intensity to 963/1022/1088 cm^−1^ peak intensity, indicating the increase in curdlan content in composites in comparison with pure HAP (**a**). Representative stress–strain curves (**b**), images of composites before (upper row) and after (lower row), the compression (**c**), and mechanical parameters of produced composites (no significant differences were noted; *p* < 0.05) (**d**).

**Figure 2 ijms-24-17178-f002:**
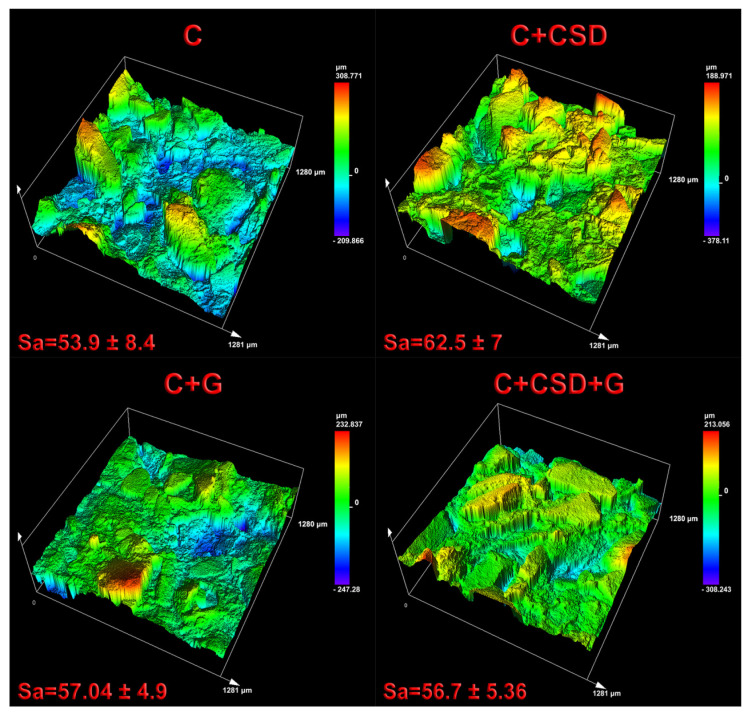
Surface topography of the synthesized biomaterials visualized by Laser Scanning Microscopy analysis.

**Figure 3 ijms-24-17178-f003:**
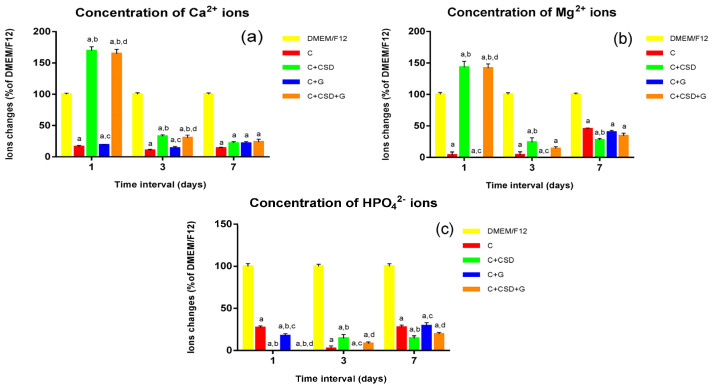
Calcium (**a**), magnesium (**b**), and phosphate (**c**) ion concentrations in DMEM/F12 medium incubated with tested composites over 1 week. *p* < 0.05, statistically significant results according to one-way ANOVA with post hoc Tukey test (a—differences in comparison with DMEM/F12; b—compared to C; c—compared to C + CSD; d—compared to C + G).

**Figure 4 ijms-24-17178-f004:**
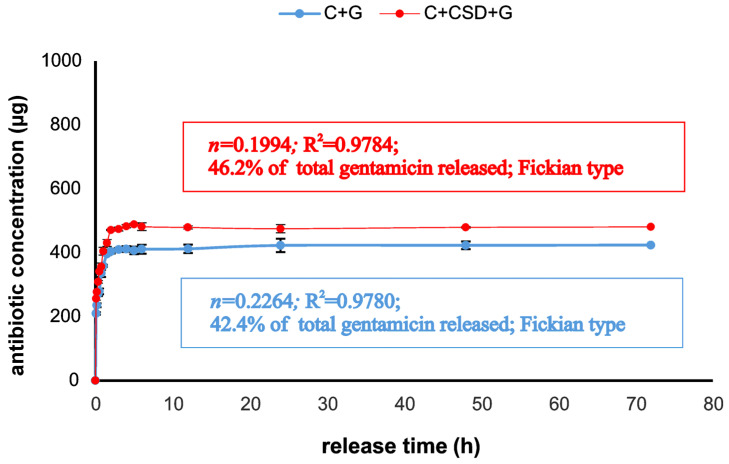
Profiles of gentamicin release from tested composites. In frames: parameters of drug release.

**Figure 5 ijms-24-17178-f005:**
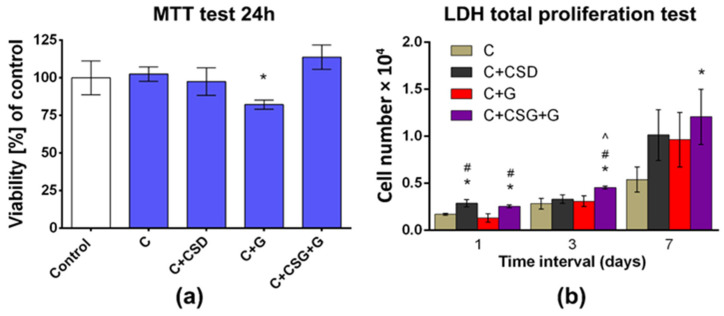
Biocompatibility evaluation of produced biomaterials with the use of a mouse primary calvarial preosteoblast cell line (MC3T3-E1): (**a**) cytotoxicity assay conducted with the use of biomaterial extracts according to ISO10993-5 procedure (control—cells treated for 24 h with cell culture medium); (**b**) MC3T3-E1 proliferation on the biomaterials assessed after 1, 3, and 7 days using total LDH test. * Statistically significant results compared to the curdlan (C) sample; # statistically significant results compared to the C + G sample; ^ statistically significant results compared to the C + CSD sample.

**Figure 6 ijms-24-17178-f006:**
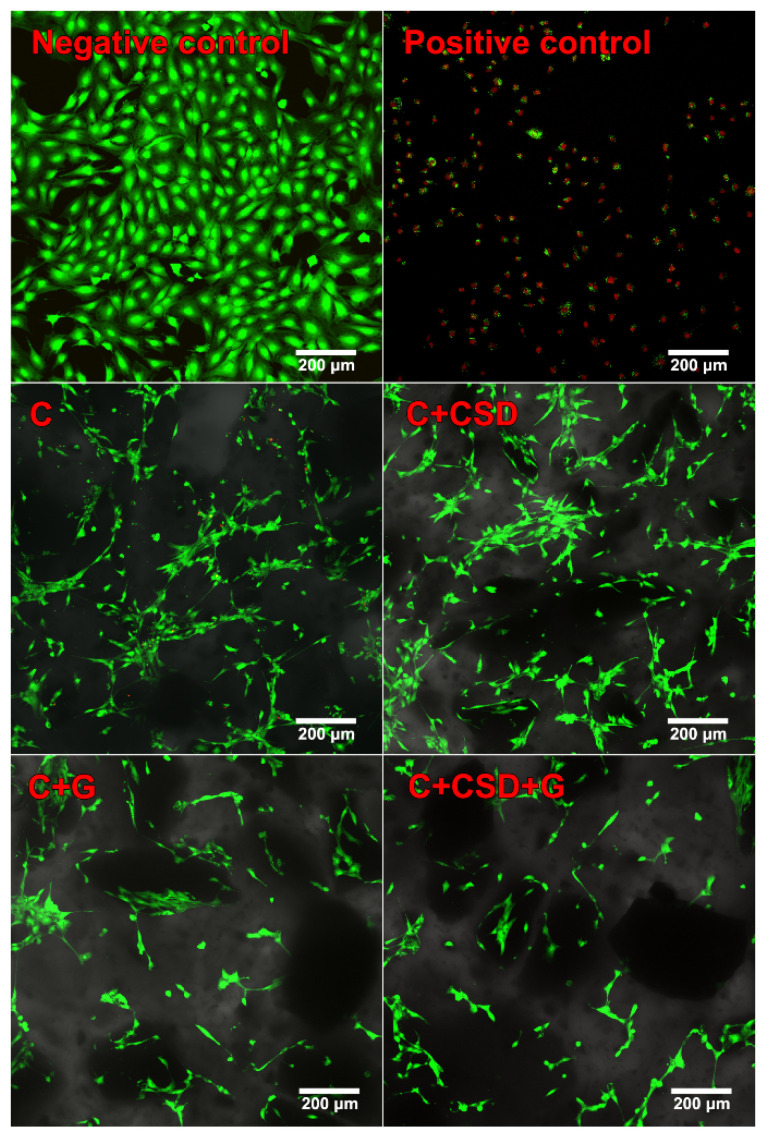
Cytotoxicity evaluation of produced biomaterials assessed by live/dead staining after 48 h of cell culture (negative control—cell culture in polystyrene wells; positive control—cells exposed for 5 h to 0.1% phenol solution; green fluorescence—live cells, red fluorescence—dead cells).

**Figure 7 ijms-24-17178-f007:**
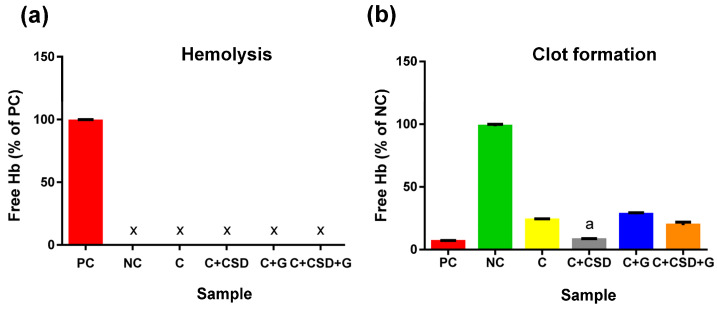
Hemolysis (**a**) and clot formation (**b**) of blood incubated with tested composite. ^x^ for hemolysis indicates the lack of statistical difference between samples and negative control, while ^a^ for clot formation indicates the lack of statistical difference between samples and positive control.

**Figure 8 ijms-24-17178-f008:**
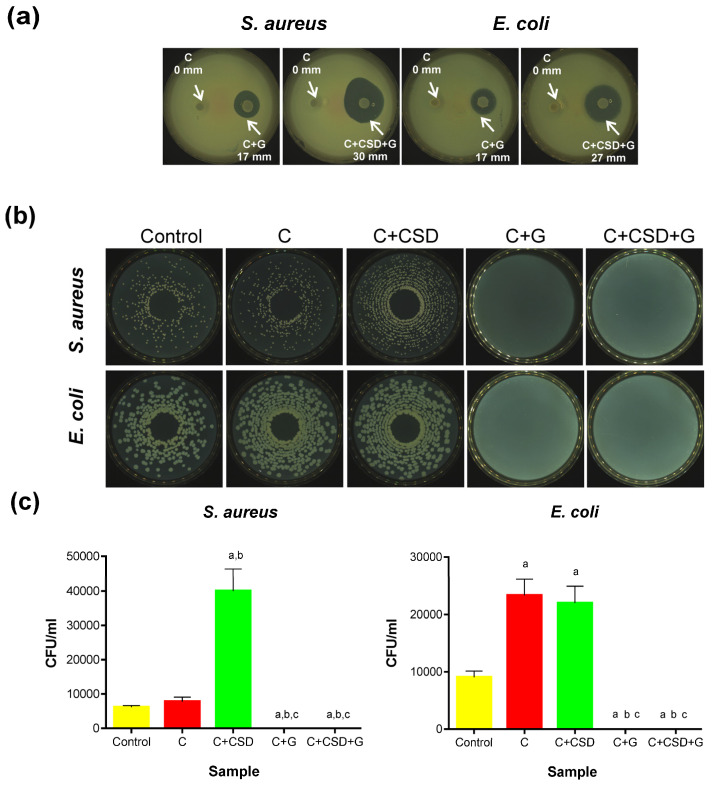
Bacterial growth inhibition in agar plate test (**a**) and results of antibacterial activity test presented as colonies growing on agar plates (**b**) and as drawings of counted live colonies (**c**). Statistically significant results according to one-way ANOVA with post hoc Tukey test (a—differences in comparison with control; b—differences compared to C; c— differences compared to C + CSD). No statistical differences were found between C + CSD and C + CSD + G.

**Figure 9 ijms-24-17178-f009:**
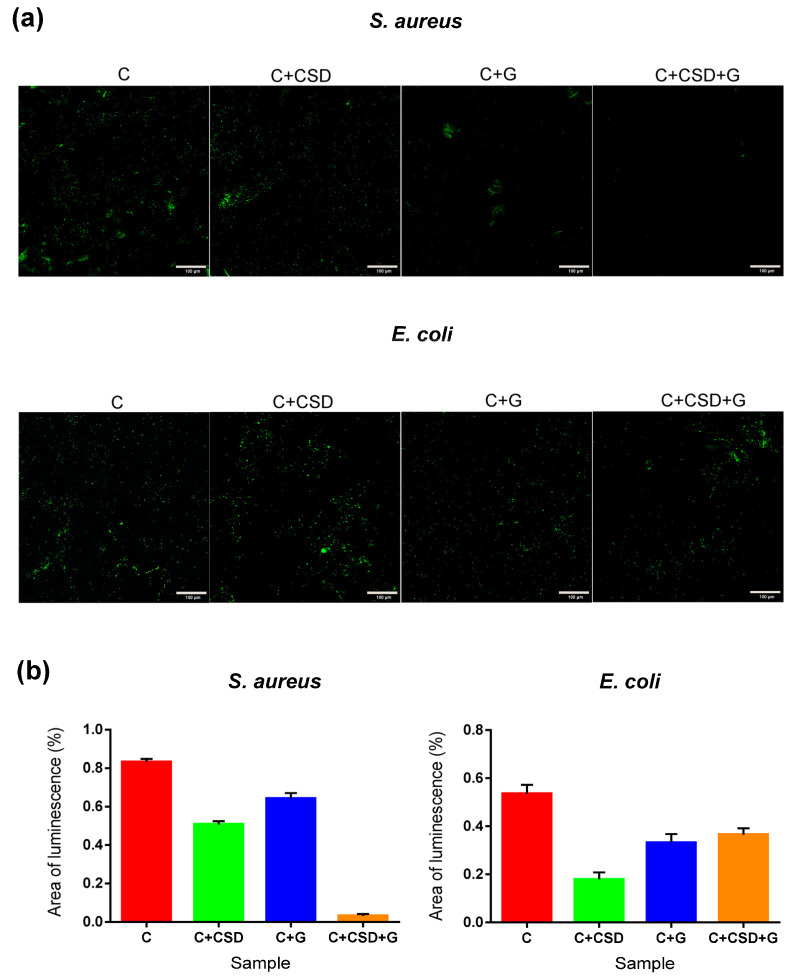
Bacterial growth adhesion as CLSM confocal images (**a**) and areas of green luminescence on representative composite images (**b**) (all results were statistically different; *p* < 0.05). The red bar indicates 50 µm.

**Table 1 ijms-24-17178-t001:** Composite samples codes and composition. C—composite (curdlan + HA); CSD—calcium sulfate dehydrate; G—gentamicin.

Sample Code	% of CSD in Ceramics Content	Amount of Compound (g)			Amount of Liquid (mL)	
		CSD Particles (0.1–0.2 mm)	HA Particles (≤0.4 mm)	Curdlan	DI H_2_O	Gentamicin (10 mg/mL)
C	0	-	8.8	3	20	-
C + CSD	1.75	0.154	8.646	3	20	-
C + G	0	-	8.8	3	18.82	1.18
C + CSD + G	1.75	0.154	8.646	3	18.82	1.18

## Data Availability

Data are contained within the article.

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
