# Peer review of "Gypsum-Related Impact on Antibiotic-Loaded Composite Based on Highly Porous Hydroxyapatite—Advantages and Disadvantages"

_ijms, 2023, doi:10.3390/ijms242417178_

Round 1

Reviewer 1 Report

Comments and Suggestions for Authors

Highly porous hydroxyapatite is at times deemed toxic and ineffective for bone regeneration due to its high adsorption of calcium and phosphate ions. This hinders osteoblast growth, creating a false impression of cytotoxicity. A recent study has shown that the addition of dihydrated calcium sulfate (CSD) compensates for this adsorption without altering other properties. This study assessed the potential of these CSD-enriched biomaterials as carriers for antibiotics. The results demonstrated a temporary compensation of calcium ions and an improvement in antibacterial activity without impacting osteoblast growth, hemocompatibility, and mechanical properties. Thus, supplementation with CSD and antibiotics could enhance hydroxyapatite-based bone regeneration biomaterials.

1) Hydroxyapatite is less soluble in water than dihydrated calcium sulfate (gypsum). This difference in solubility is crucial concerning the stability and reactivity of these materials in various contexts, particularly in the field of bone regeneration. The exclusive use of gypsum is recommended to assess their effects.

2) The difference in uncertainty between Figure 7 and Figure 8 may stem from the nature of the data or measurements involved. Figure 7 possibly represents more direct, precise, or reproducible data, thereby reducing associated uncertainty. On the other hand, Figure 8 may involve more complex measurements or variables prone to variations or unpredictable factors, introducing a degree of uncertainty into the results. The presence or absence of uncertainty could be linked to the nature of the collected data, the methodology employed, or specific experimental conditions.

3) To ascertain the presence of an insertion or doping, it is crucial to characterize the final product.

4) According to our findings, gypsum-enriched composites designed for bone regeneration can act as a matrix for the release of antibiotics at bone defect sites. The results indicate that the addition of 1.75% gypsum and gentamicin led to a short-term compensation of calcium ions in environments incubated with the composite. The combination of both additives also enhanced antibacterial activity against representative strains of both Gram-positive and Gram-negative bacteria associated with bone infections, without inducing negative effects on cell proliferation, hemocompatibility, and mechanical properties. Thus, supplementation with gypsum and antibiotics could provide advanced functionalities to hydroxyapatite-based bone regeneration materials, characterized by a large surface area and increased Ca+2 adsorption capacity. Propose an explanatory mechanism for these effects.

5) The conclusion is very brief, preferably to develop.

6) References should be recent and pertinent to the field of study.

Comments on the Quality of English Language

 Minor editing of English language required

Author Response

Reviewer 1

Highly porous hydroxyapatite is at times deemed toxic and ineffective for bone regeneration due to its high adsorption of calcium and phosphate ions. This hinders osteoblast growth, creating a false impression of cytotoxicity. A recent study has shown that the addition of dihydrated calcium sulfate (CSD) compensates for this adsorption without altering other properties. This study assessed the potential of these CSD-enriched biomaterials as carriers for antibiotics. The results demonstrated a temporary compensation of calcium ions and an improvement in antibacterial activity without impacting osteoblast growth, hemocompatibility, and mechanical properties. Thus, supplementation with CSD and antibiotics could enhance hydroxyapatite-based bone regeneration biomaterials.

Q1) Hydroxyapatite is less soluble in water than dihydrated calcium sulfate (gypsum). This difference in solubility is crucial concerning the stability and reactivity of these materials in various contexts, particularly in the field of bone regeneration. The exclusive use of gypsum is recommended to assess their effects.

A1) Thank you for your comment. Indeed, hydroxyapatite is much less soluble than gypsum. This was the reason why we used gypsum as an additive to HAp for calcium ions compensation in our previous study entitled “Gypsum-related compensation of ions uptake by highly porous hydroxyapatite ceramics – Consequences for osteoblasts growth and proliferation” Biomaterials Advances 133 (2022) 112665. We agree with the reviewer's observation that dissolving the plaster may change the structure of the biomaterial, increasing its porosity and weakening its mechanical properties. For this reason, in the previous study, we assessed both the change in structure (by SEM), porosity (by microCT and mercury porosimetry), and mechanical properties of the composite after incubation in SBF after dissolution of the gypsum phase. The study showed that dissolution of the gypsum did not induce a detectable change in the porosity, structure, or mechanical parameters of the high-porosity HAP-based composite. This was in line with our expectations, because the gypsum content in the composite, selected as optimal for the ion uptake compensation process, was low (less than 2%). In this study, we did not repeat this effect to avoid duplication of research. The gentamicin-enriched composite contained a very low drug concentration (1 mg/g dry weight), which made it not significantly different from the gentamicin-free composite described in the previous work.

Q2) The difference in uncertainty between Figure 7 and Figure 8 may stem from the nature of the data or measurements involved. Figure 7 possibly represents more direct, precise, or reproducible data, thereby reducing associated uncertainty. On the other hand, Figure 8 may involve more complex measurements or variables prone to variations or unpredictable factors, introducing a degree of uncertainty into the results. The presence or absence of uncertainty could be linked to the nature of the collected data, the methodology employed, or specific experimental conditions.

A2) We admit that the Reviewer is right. Indeed, the results shown in Figure 8 present results with a significant risk of uncertainty and error. In particular, growth inhibition zones may be difficult to precisely measure due to their irregular shape (the zone is not a perfect circle). For this reason, we treat the results of this test (shown in Figure 8a) as pilot studies (as noted in the text). The results of the microbiological activity test presented in Figures 8b-c are much more reliable. Photos of Petri plates containing bacterial colonies indeed provide only illustrative knowledge, but Figure 8c shows quantitative calculations of CFU on these plates, counted by the automatic colony reader Scan 300 (equipped with the ... program) on the plates. These results have a much lower risk of inaccuracy and subjectivity.

Q3) To ascertain the presence of an insertion or doping, it is crucial to characterize the final product.

A3) Gentamicin was introduced into the composite formula by adding a known dose of the drug to the composite mass before the polymerization stage. Polymerization completed the composite production process. It was not washed or subjected to any other post-production treatment, which ensured that the full dose of the drug introduced into the biomaterial was maintained. The final product was characterized to determine its properties. First, the method chosen to confirm the gentamicin content in the composite was the FTIR technique. The FTIR spectrum did not show the presence of gentamicin in the composite, which can be easily explained by the low drug content in the biomaterial (1 mg drug/1 g dry weight of the composite, which is equivalent to 0.1% content). However, the second method of characterization of the composite, namely the detection of substances released from the biomaterial during incubation in PBS, showed that gentamicin was released from the composite in an amount of approximately 45% of the theoretical content. This confirms that it has been effectively introduced into the biomaterial formula. The third evidence for the successful introduction of gentamycin into the produced biomaterials is the antibacterial activity demonstrated by biomaterials containing the drug, while it was not detected by the reference biomaterials not containing the drug.

Q4) According to our findings, gypsum-enriched composites designed for bone regeneration can act as a matrix for the release of antibiotics at bone defect sites. The results indicate that the addition of 1.75% gypsum and gentamicin led to a short-term compensation of calcium ions in environments incubated with the composite. The combination of both additives also enhanced antibacterial activity against representative strains of both Gram-positive and Gram-negative bacteria associated with bone infections, without inducing negative effects on cell proliferation, hemocompatibility, and mechanical properties. Thus, supplementation with gypsum and antibiotics could provide advanced functionalities to hydroxyapatite-based bone regeneration materials, characterized by a large surface area and increased Ca+2 adsorption capacity. Propose an explanatory mechanism for these effects.

A4) We added a paragraph to the results and Discussion section, explaining our hypothesis on the mechanism of these actions (highlighted in light grey in the manuscript):

“In our opinion, the concept of the presented design contains two basic elements that may affect various aspects of bone tissue regeneration: antibiotic loading and gypsum supplementation. The porous formula of the composite allows it to absorb water from surrounding liquid. This results in the gradual dissolution of the gypsum phase (1.75% of total composite weight), releasing calcium ions. This compensates for the adsorption of calcium and phosphate ions from the medium driven by hydroxyapatite of high specific surface area and positively affects the growth and proliferation of osteoblasts, especially in the first adaptive phase of cell-biomaterial contact. Gentamicin undergoes the release from the matrix, inhibiting the growth of bacteria. It seems that the rate of drug release is facilitated by gypsum dissolution accompanied by the formation of additional pores within the composite. Drug release is higher (Figure 4) and zones of bacterial growth inhibition are larger (Figure 8a) in the case of gypsum-and-antibiotic-loaded material in comparison to antibiotics-loaded one. Although evaluation of the bacterial-killing potential of the materials suggested that calcium ions released from gypsum may somewhat stimulate bacterial growth, it seems that simultaneous loading with antibiotics eliminates this effect (Figure 8b-c). Notably, these additives are neutral for mechanical properties and the 3D structure of final composites (probably due to their low content). Overall, it seems clear that low doses of CSD and antibiotic, added to high-specific-area hydroxyapatite-based biomaterials, may be a promising tool for application of this type of ceramics in the biomedical field.”

Q5) The conclusion is very brief, preferably to develop.

A5) We tried to develop the conclusions by adding some notes and observations. The new Conclusions are as below:

“According to our results, gypsum-enriched composites designed for bone tissue regeneration can serve as a matrix for antibiotics delivery to bone defect sites. The results showed that the addition of 1.75% gypsum and gentamicin caused short-term calcium ions compensation in media incubated with the composite. The combination of both additives also increased antibacterial activity for Gram-positive and Gram-negative strains representative of bone infections without negative effects on cell proliferation, hemocompatibility, and mechanical parameters. The properties of these two additives likely cooperate in the final effect on osteoblasts and bacteria. According to our observations, the dissolving gypsum may not only release calcium ions which compensate for their uptake from the environment by high-specific-area hydroxyapatite. It may also facilitate the action of antibiotics by pores generation in bio-material and enhancement of liquid circulation within its structure. Thus, gypsum and antibiotic supplementation may provide advanced functionality for bone regeneration materials based on hydroxyapatite of high surface area and increased high Ca+2 sorption capacity.”

Q6) References should be recent and pertinent to the field of study.

A6) Thank you for this not. Some new references were inserted into the text (highlighted in light grey in the manuscript and in references section).

Q7) Comments on the Quality of English Language:  Minor editing of English language required.

A7) We corrected the text in cooperation with the skilled person.

Reviewer 2 Report

Comments and Suggestions for Authors

The study investigated if the hydroxyapatite supplemented with calcium sulfate dihydrate (CSD) could be an antibiotic carriers. Tests on material properties, drug release, cell compatibility, and antibacterial effects were conducted. Results showed that including small amounts of gypsum and gentamicin balanced calcium ions temporarily, enhancing antibacterial properties against bone infection bacteria without compromising bone cell growth, blood compatibility, or material strength. This suggests that combining gypsum and antibiotics could improve materials for bone regeneration, especially those using high surface area hydroxyapatite with increased calcium ion absorption.

Major comments:

1. Figure 6, the density and morphology of cells in control are quite different from all other samples, and specially for group of C+CSD+G. It's hard to tell if the cells were in health status. Can author provide more expatiation? Also, there is few red fluorescent cells in the images, do authors have any control with all dead cells labeled by red fluorescence?

2. Figure 9, a) the images are too dim, do author have higher resolution images? b) please add the S.D. and p values in Figure 9b. 

3. Why does adding gentamicin increase the adhesion of E. coli (for C+CSD+G)?

4. How did author do the quantification based on luminescence (Figure 9b)?

Minor comments:

1. Figure 1b, the image is very dim, do authors have more clear images?

2. Figure 4, can author add S.D. for each date point?

3. Figure 5, a) Why the average viability of C+CSG+G is much higher than 100%? b) I suggest using log scale for y-axis to show cells number. Do authors have controls?

Author Response

Reviewer 2

The study investigated if the hydroxyapatite supplemented with calcium sulfate dihydrate (CSD) could be an antibiotic carriers. Tests on material properties, drug release, cell compatibility, and antibacterial effects were conducted. Results showed that including small amounts of gypsum and gentamicin balanced calcium ions temporarily, enhancing antibacterial properties against bone infection bacteria without compromising bone cell growth, blood compatibility, or material strength. This suggests that combining gypsum and antibiotics could improve materials for bone regeneration, especially those using high surface area hydroxyapatite with increased calcium ion absorption.

Major comments:

Q1) Figure 6, the density and morphology of cells in control are quite different from all other samples, and specially for group of C+CSD+G. It's hard to tell if the cells were in health status. Can author provide more expatiation? Also, there is few red fluorescent cells in the images, do authors have any control with all dead cells labeled by red fluorescence?

A1) Thank you for this comment. Figure 6 represents results obtained by the Live-dead staining assay. The cytotoxicity was assessed directly (qualitatively) by culturing cells on the tested biomaterials for 2 days after which cells were stained with calcein-AM and propidine iodide (PI) dyes and visualized using CLSM. The calcein generated from Calcein-AM by esterase in a viable cell emits a strong green fluorescence. Therefore, calcein-AM only stains viable cells. Alternatively, the nuclei staining dye PI cannot pass through a viable cell membrane. It reaches the nucleus by passing through disordered areas of dead cell membrane, and intercalates with the DNA double helix of the cell to emit red fluorescence. The conducted test allows for analysis of only the cytotoxicity of tested samples. Where lack of emitted red color and the presence of green color means non-toxicity of tested biomaterials. The control cells were cultured in polystyrene wells on a tissue culture-treated plate that is characterized by an appropriate surface for cell attachment, growth and differentiation. Due to this fact, it is normal to obtain some differences in the cell density and morphology between tested samples and control. Nevertheless, to be able to assess cell morphology, it is necessary to stain the cytoskeleton and cell nuclei with fluorescent stainings (e.g. DAPI, Phalloidin) that are dedicated to that purpose. An additional confirmation of the health of the cells and the lack of toxicity of the biomaterials is the increase in the number of cells over time in the proliferation test. Also positive control image was added to the figure. Now the image is as follows:

(please find the figure in the enclosed file)

Figure legend is now as follows:

Figure 6. Cytotoxicity evaluation of produced biomaterials assessed by Live/Dead staining after 48h of cell culture (Negative control –cell culture in polystyrene wells; Positive control – cells exposed for 5h to 0.1% phenol solution; green fluorescence – live cells, red fluorescence – dead cells).

Q2) Figure 9, a) the images are too dim, do author have higher resolution images? b) please add the S.D. and p values in Figure 9b.

A2) Thank you for this remark. We tried to light the images up a bit and added SD values to Figure 9b. All results were statistically different (p<0.05).

Q3) Why does adding gentamicin increase the adhesion of E. coli (for C+CSD+G)?

A3) We admit that the explanation of this phenomenon is not easy to find. Especially since this observation did not appear for S. aureus adhesion (in this case, the significant reduction of bacterial adhesion was observed for C+CSD+G). We tried to comment on this observation in the text, as follows: “For E. coli, in contrast, the effect of gentamicin is less significant; moreover, no cumulative effect of gypsum and drug was observed. In general, the susceptibility of E. coli and S. aureus to gentamicin was comparable because gentamicin MICs for these two strains are similar [34,35]. Also, the growth inhibition zones around composite with CSD and drug (C+CSD+G) were smaller for E. coli than for S. aureus (Figure 8a). These observations may suggest that tested drug-loaded composite is more effective against Gram-positive than Gram-negative bacteria.”

Another explanation may lie in the fact that calcium ions released from CSD stimulated bacterial growth. It can be suggested by a higher number of surviving bacteria incubated with C+CSD material in comparison with control (Figure 8c, for both strains). Therefore, calcium ions could support the viability of E. coli in adhesion tests, thus increasing their potential for adhesion to solid surfaces. However, we still cannot firmly explain why this phenomenon did not appear for S. aureus, only for E. coli. As already mentioned, this difference may come from different properties of Gram-positive than Gram-negative bacteria.

Q4) How did author do the quantification based on luminescence (Figure 9b)?

A4) For this purpose, we used the ImageJ program. First, we adjusted the color balance (in the same manner and to the same degree for all images), to reduce the grey luminescence of the background. Then we analyzed the histograms of color intensity for all images.  In general, the adhesion of bacteria was low for all tested materials (less than 1% of the total image area was covered by lightning bacteria). For this reason, the images were dim, we apologize for this, but intensification of laser power caused also the increase of background luminescence and enhanced the fading of the bacteria's glow due to the decomposition of the dye used for bacterial detection.

Minor comments:

Q1) Figure 1b, the image is very dim, do authors have more clear images?

A1) Thank you for your note, were tried to bright up this image. Figure 1 is now as follows:

(please find the figure in the enclosed file)

Q2) Figure 4, can author add S.D. for each date point?

A2) Thank you for your remark, SD values were added to the figure 4.

Q3) Figure 5, a) Why the average viability of C+CSG+G is much higher than 100%? b) I suggest using log scale for y-axis to show cells number. Do authors have controls?

A3) Thank you for this remark. The  MTT  test  can  estimate  cell  metabolic  activity  by the  tetrazolium  intracellular  reduction  and  formazan formation, which is proportional to the number of viable cells with active mitochondrial dehydrogenases. The increase in viability above 100% in the MTT test can be explained by the positive impact of biomaterial extracts on cell metabolism. Presenting results as a viability percentage of non-toxic control is due to the ISO 10993-5:2009 standard, which defines materials as toxic if the viability is lower than 70%.

Reviewer 3 Report

Comments and Suggestions for Authors

In this study, CSD-HA ceramics composites were prepared and characterized. Some issues need to be addressed before the manuscript is accepted.

Introduction

1. Line 39, “… is related to increased bio-mineralization after the implantation.” Did you mean bio-mineralization formation at the implant surface? It is because “the negative one concerns the reduced ions availability for cells that are responsible for bone tissue formations.”. Did you mean the Ca ions release to the surrounding cells?  

Materials and Methods

1. The ceramics powders were ground in a mortar and sieved to obtain granules of the size ≤ 0.4 mm. The particle size may vary from several um to several hundred um. Did you evaluate the mean particle size by, e.g., dynamic light scattering particle analyser or TEM?   

2. Statistical analysis, line 514, significant differences between the test groups p < 0.5?

3. Did you examine the osteoblast mineralization by Alizarin red S assay to compare between the four groups?

Results and Discussion

1. In Table 1d, any significant differences in compression and compressive strength between groups? Please indicate the significant difference between groups using capital or small letters, etc.

2. In Fig. 6, the green fluorescence is clear but it is not clear to observe the red fluorescence.

3. In Fig. 9 a, could you improve the CLSM images? It is not clear. 

4. The incorporation of CSD in HA ceramics would enhance uptake of Ca and Mg ions. Would this affect the bone healing/regeneration at early stage after implantation as the Ca and Mg ions release is probably very slow.    

Comments on the Quality of English Language

Minor English is recommended.

Author Response

Reviewer 3

In this study, CSD-HA ceramics composites were prepared and characterized. Some issues need to be addressed before the manuscript is accepted.

Introduction

Q1) Line 39, “… is related to increased bio-mineralization after the implantation.” Did you mean bio-mineralization formation at the implant surface? It is because “the negative one concerns the reduced ions availability for cells that are responsible for bone tissue formations.”. Did you mean the Ca ions release to the surrounding cells? 

A1) Thank you for Your comment, now we realize that the quoted fragment was not understandable enough for the Reader. We tried to edit this fragment in the Introduction which is now cited below (and marked in turquoise in the text):

“This affects bone tissue regeneration in both positive and negative manner. The positive aspect of this phenomenon is related to increased bio-mineralization after the implantation due to hydroxyapatite bioactivity. Hydroxyapatite intensely adsorbs calcium and phosphate ions from surrounding liquid which leads to the formation of biological apatite in the ceramics formation and the rate of this process depends on the specific surface area of ceramics (thus with the number of nucleation sites). However, the disappearance of calcium and phosphate ions leads to negative consequences because the reduced ions availability for cells that are responsible for bone tissue formation can limit the osteoblasts proliferation and differentiation [3].”

Materials and Methods

Q1) The ceramics powders were ground in a mortar and sieved to obtain granules of the size ≤ 0.4 mm. The particle size may vary from several um to several hundred um. Did you evaluate the mean particle size by, e.g., dynamic light scattering particle analyser or TEM?  

A1) No, the size of granules was not confirmed by tools like TEM or dynamic light scattering particle analyzer. We relied on the dimensions of sieve holes. The granules and curdlan served as a matrix enabling the introduction of CSD and gentamicin and the effect of the latter was in focus of our interest. For this reason, the hydroxyapatite-curdlan matrix (named “C”) was used as a reference material in all experiments, to take into account the effect of hydroxyapatite alone.

Q2) Statistical analysis, line 514, significant differences between the test groups p < 0.5?

A2) The significance level was considered at p < 0.05. The results were expressed as the mean values ± standard deviation (SD). It was added to the text and marked in turquoise.

Q3) Did you examine the osteoblast mineralization by Alizarin red S assay to compare between the four groups?

A3) Thank you for this comment. Comprehensive biological evaluation (e.g. osteoblasts mineralization) is planned to be carried out in the next part of the research.

Results and Discussion

Q1) In Table 1d, any significant differences in compression and compressive strength between groups? Please indicate the significant difference between groups using capital or small letters, etc.

A1) Thank you for this remark. No statistical differences were noted for data presented in Figure 1d. It was mentioned in Figure 1 legend.

Q2) In Fig. 6, the green fluorescence is clear but it is not clear to observe the red fluorescence.

A2) Thank you for this remark. The poor visibility of red fluorescence is caused by the high biocompatibility and non-toxicity of the produced biomaterials, and also by partial coverage of the green and red signals.

Q3) In Fig. 9 a, could you improve the CLSM images? It is not clear.

A3) The images were brightened up a bit, thank you for this remark.

Q4) The incorporation of CSD in HA ceramics would enhance uptake of Ca and Mg ions. Would this affect the bone healing/regeneration at early stage after implantation as the Ca and Mg ions release is probably very slow.   

A4) This comment is very appropriate. As was explained in our earlier study, despite a short-time burst of calcium ions due to the CSD presence, preosteoblast cells cultured in the presence of gypsum-supplemented composites showed an increased proliferation rate in comparison with the cells cultured on the composite without gypsum additive. It should be noted that calcium and magnesium ions are crucial for preosteoblasts' adhesion to biomaterial during the adaptation phase (Zalewska et al., Biomaterials Advances 133 (2022) 112665). After the adaptation, the cells adhered to the surface of biomaterial continuing their proliferation of differentiation, and are less sensitive to the deficit of the mentioned ions. In this way, the task of accelerating the initial phase of regeneration would be achieved. However, we agree that maintaining this effect for a longer period would probably be even more beneficial for the regeneration process of bone tissue. This allows us to continue our research to extend the observed effect of calcium ion release for longer than a few days.

  1. Q) Comments on the Quality of English Language: Minor English is recommended.
  2. A) We corrected the text in cooperation with skilled person.

Round 2

Reviewer 1 Report

Comments and Suggestions for Authors

Acceptable for publication.

Reviewer 3 Report

Comments and Suggestions for Authors

No further comments. The authors addressed my comments.